# Iron overload inhibits BMP/SMAD and IL-6/STAT3 signaling to hepcidin in cultured hepatocytes

**Edouard Charlebois, Kostas Pantopoulos** \*

Lady Davis Institute for Medical Research, Jewish General Hospital and Department of Medicine, McGill University, Montreal, Quebec, Canada

\* kostas.pantopoulos@mcgill.ca

## Abstract

Hepcidin is a peptide hormone that targets the iron exporter ferroportin, thereby limiting iron entry into the bloodstream. It is generated in hepatocytes mainly in response to increased body iron stores or inflammatory cues. Iron stimulates expression of bone morphogenetic protein 6 (BMP6) from liver sinusoidal endothelial cells, which in turn binds to BMP receptors on hepatocytes and induces the SMAD signaling cascade for transcriptional activation of the hepcidin-encoding *HAMP* mRNA. SMAD signaling is also essential for inflammatory *HAMP* mRNA induction by the IL-6/STAT3 pathway. Herein, we utilized human Huh7 hepatoma cells and primary murine hepatocytes to assess the effects of iron perturbations on signaling to hepcidin. Iron chelation appeared to slightly impair signaling to hepcidin. Subsequent iron supplementation not only failed to reverse these effects, but drastically reduced basal *HAMP* mRNA and inhibited *HAMP* mRNA induction by BMP6 and/or IL-6. Thus, treatment of cells with excess iron inhibited basal and BMP6-mediated SMAD5 phosphorylation and induction of *HAMP*, *ID1* and *SMAD7* mRNAs in a dose-dependent manner. Iron also inhibited IL-6-mediated STAT3 phosphorylation and induction of *HAMP* and *SOCS3* mRNAs. These responses were accompanied by induction of *GCLC* and *HMOX1* mRNAs, known markers of oxidative stress. We conclude that hepatocellular iron overload suppresses hepcidin by inhibiting the SMAD and STAT3 signaling pathways downstream of their respective ligands.

## Introduction

Hepcidin is a hormonal regulator of systemic iron traffic [1, 2]. It is highly expressed in hepatocytes as pre-pro-hepcidin, which undergoes proteolytic processing to yield a mature peptide of 25 amino acids. Hepcidin operates by binding to the iron exporter ferroportin in target cells, such as tissue macrophages, duodenal enterocytes and hepatocytes. The binding of hepcidin directly inhibits iron efflux from ferroportin into plasma, and also targets ferroportin for lysosomal degradation. Hepcidin-mediated inhibition of iron entry into plasma serves as a

**Data Availability Statement:** All relevant data are within the manuscript and its Supporting information files.

**Funding:** This work was supported by a grant from the Canadian Institutes of Health Research (CIHR;

PJT-159730). EC was funded by a fellowship from the Natural Sciences and Engineering Research Council of Canada (NSERC) and is currently a recipient of a fellowship from the Fonds de recherche du Québec – Santé (FRQS).

**Competing interests:** The authors have declared that no competing interests exist.

homeostatic response to increased body iron stores, or as an innate immune response to infection [3, 4].

Expression of hepcidin-encoding *HAMP* mRNA is primarily induced in response to iron or inflammatory signals. The underlying mechanisms are fairly well characterized; however, critical aspects of iron sensing remain elusive [2, 5]. Hepatic iron loading triggers expression and release of bone morphogenetic protein 6 (BMP6) from liver sinusoidal endothelial cells [6]; these cells also constitutively express BMP2 [7]. BMP6 binds as homodimer to ALK2 and as heterodimer with BMP2 to ALK3 [8], which are type I BMP receptors on hepatocytes. Binding of the ligand activates SMAD signaling via a mechanism that is enhanced by hemojuvelin (HJV), a BMP co-receptor and other auxiliary factors. The signaling cascade involves phosphorylation of SMAD1/5/8, recruitment of SMAD4 and formation of a complex that translocates to the nucleus for binding to the *HAMP* promoter and activation of *HAMP* mRNA transcription.

The inflammatory cytokine IL-6 is another potent inducer of hepcidin. The binding of IL-6 to its receptor on hepatocytes triggers phosphorylation of STAT3 by JAK1/2 kinases, which in turn translocates to the nucleus to transcriptionally induce hepcidin [9, 10]. It is well established that this pathway requires active BMP/SMAD signaling [11, 12].

Treatment of hepatoma cells or primary murine hepatocytes with BMP6, BMP2 or IL-6 recapitulates hepcidin induction [8, 12–14]. However, while dietary, genetic or pharmacological iron loading readily induce hepcidin expression in vivo [15–17], exposure of cells to iron-loaded transferrin or iron salts was shown to rather suppress hepcidin [14, 18, 19]. Herein, we explore the mechanisms by which iron affects hepcidin expression in cultured cells. We demonstrate that iron loading suppresses *HAMP* mRNA levels by inhibiting basal SMAD signaling. Moreover, excess iron inhibits ligand-induced SMAD and STAT3 signaling and antagonizes *HAMP* mRNA induction by BMP6 or IL-6.

## Materials and methods

### Cell culture

Human Huh7 hepatoma cells were cultured in Dulbecco's modified Eagle's medium supplemented with 10% heat-inactivated fetal bovine serum (Wisent), non-essential amino acids, 100 U/ml penicillin and 100 μg/ml streptomycin. Primary hepatocytes were isolated from livers of C57BL/6 wild type mice (22–25 g) by using a two-step collagenase perfusion technique as previously described [12]. The experimental procedure was performed at the animal facility of the Lady Davis Institute for Medical Research and was approved by the Animal Care Committee of McGill University (protocol 4966). The cells were seeded onto tissue culture dishes (26,000 cells/cm$^2$) in Williams' medium E (Gibco) supplemented with 10% heat-inactivated fetal bovine serum, 2 mM L-glutamine, 100 U/ml penicillin, 100 μg/ml streptomycin, and allowed 90 min to attach before washing and overnight culture. The serum-containing medium was removed the next morning, and the cells were subjected to different culture conditions in serum-free Williams' medium E. In control groups, cells were incubated with medium alone for the indicated time of the experiment.

### Materials

Deferoxamine (DFO), FeSO$_4$ and ferric ammonium citrate (FAC) were purchased from Sigma-Aldrich. Following recombinant proteins were used: murine IL-6 (#5216, Cell Signaling), human IL-6 (#I1395, Sigma-Aldrich), human BMP6 (#507-BP, R&D Systems), and human holo-transferrin (#T4132, Sigma). Salicylaldehyde isonicotinoyl hydrazone (SIH) was a

gift of the late Dr. Prem Ponka. Fe-SIH was prepared by mixing SIH with ferric citrate in 2:1 ratio [20].

## Quantitative real-time PCR (qPCR)

The cells were washed twice with phosphate-buffered-saline and RNA was extracted by using the RNeasy kit (Qiagen). cDNA was synthesized from 1 μg RNA by using the OneScript cDNA Synthesis Kit (ABM). Gene-specific primers pairs (S1 Table) were validated by dissociation curve analysis and demonstrated amplification efficiency between 90–110%. SYBR Green (Bioline) and primers were used to amplify products under following cycling conditions: initial denaturation 95˚C 10 min, 40 cycles of 95˚C 5 s, 58˚C 30 s, 72˚C 10 s, and final cycle melt analysis between 58˚-95˚C. Relative mRNA expression was calculated by the comparative Ct method. Data were normalized to human ribosomal protein RPS18 for Huh7 cells or murine Rpl19 for primary murine hepatocytes. Data are reported as fold changes compared to untreated control cells. Original Ct values can be found in S1 Dataset.

## Western blotting

Following 2x wash with phosphate-buffered-saline, the cells were lysed as previously described [21]. Cell lysates containing 30 μg of proteins were analyzed by SDS-PAGE on 10% gels and proteins were transferred onto nitrocellulose membranes (BioRad). The blots were blocked in 10% bovine serum albumin or 10% fat-free skim milk in tris-buffered saline (TBS) containing 0.1% (v/v) Tween-20 (TBS-T), and probed overnight with 1:1000-diluted (unless otherwise indicated) primary antibodies against pSMAD5 (phospho-S463/465, #92698 rabbit monoclonal; Abcam), SMAD1 (#9743 rabbit polyclonal; Cell Signaling), pSTAT3 (phospho-Y705, #9138, mouse monoclonal; Cell Signaling), STAT3 (#9139, mouse monoclonal; Cell Signaling), ferritin heavy chain (1:500-diluted, #NB600-920, Novus), or β-actin (Sigma). Following a 3x wash with TBS-T, the membranes were incubated with 1:5000-diluted anti-mouse or 1:20000-diluted anti-rabbit peroxidase-coupled secondary antibodies for 2 h. Immunoreactive bands were detected by enhanced chemiluminescence with the Western Lightning ECL Kit (Perkin Elmer). Original Western blot images can be found in S1 Raw images.

## Statistical analysis

Statistical analysis was performed by using the Prism GraphPad software (version 7.0e). Multiple groups were subjected to analysis of variance (ANOVA) with Tukey's post-test comparison (one-way ANOVA) or Sidak's multiple comparison (two-way ANOVA). A probability value $p < 0.05$ was considered statistically significant.

# Results and discussion

## Iron overload abrogates BMP/SMAD and IL-6/STAT signaling to hepcidin in Huh7 cells

We have reported that iron-depleted mice fail to mount an appropriate inflammatory induction of hepcidin and this response can be restored by iron supplementation [12]. This prompted us to evaluate the potential role of hepatocellular iron status on hepcidin regulation. Human Huh7 hepatoma cells were treated with IL-6 and/or BMP6 for 4 h following pre-treatment (or not) with the iron chelator deferoxamine (DFO) for 18 h. The conditions for IL-6 and BMP6 treatments in terms of dose and duration were optimized in previous experiments [12]. Likewise, conditions for iron chelation were based on previous literature [22, 23]. DFO did not affect IL-6-mediated induction of *HAMP* mRNA or STAT3 phosphorylation but

modestly mitigated BMP6-mediated *HAMP* mRNA induction and SMAD5 phosphorylation (Fig 1A and 1B). Moreover, it negatively affected levels of *ID1* mRNA (Fig 1C), a known target of BMP6/SMAD signaling [24]. Surprisingly, administration of ferric ammonium citrate (FAC) together with IL-6 and/or BMP6 (following DFO wash) did not restore but rather strongly suppressed *HAMP* and *ID1* mRNA induction, as well as SMAD5 phosphorylation by BMP6. In addition, FAC abolished *HAMP* mRNA induction by IL-6. FAC is widely used as an iron donor in cell culture experiments [25, 26]. The underlying iron transport pathway is not fully understood but experiments in primary rat hepatocytes suggested a facilitated diffusion mechanism [27, 28].

As expected, the DFO treatment resulted in increased expression of *TFRC* mRNA (Fig 1D), which encodes transferrin receptor 1 (TfR1), and serves as marker of iron deficiency [29]. Furthermore, FAC administration for 6 h prevented *TFRC* mRNA induction by DFO, suggesting that it terminated cellular iron deficiency. Notably, under these experimental conditions, ferritin expression remained suppressed (Fig 1B), in spite of excess iron. Increased ferritin steady-state levels are only evident after longer (48 h) treatments with FAC (S1 Fig). Considering that ferritin operates by storing and detoxifying excess iron [30], the lack of ferritin induction after a relatively short (6 h) treatment with FAC (Fig 1B and S1 Fig), and the reported rapid kinetics ($25 \times 10^6$ atoms/cell/min) of FAC uptake by hepatocytes [27] are consistent with accumulation of redox-active "free" iron in the cells that can trigger oxidative stress [31]. In support to this interpretation, the FAC treatment increased expression of *GCLC* mRNA (Fig 1E), an oxidative stress marker that encodes glutamate-cysteine ligase catalytic subunit [32, 33]. Moreover, FAC also induced expression of the heme oxygenase 1-encoding *HMOX1* mRNA (Fig 1F), another established marker of oxidative stress [33, 34]. The above data suggest that hepatocellular iron overload acts as a potent inhibitor of *HAMP* mRNA induction by the BMP6/SMAD and also IL-6/STAT signaling pathways.

The inhibitory effects of FAC could be reproduced with $FeSO_4$, an additional source of inorganic iron (Fig 2A). Holo-transferrin was less efficient in inhibiting BMP6-mediated *HAMP* mRNA induction (Fig 2A), possibly because it elicits modest iron loading compared to inorganic iron salts. Similar results with FAC and $FeSO_4$ were obtained by using the iron donor Fe-SIH (Fig 2B), which is generated by mixing ferric citrate with the iron chelator SIH [20]. Notably, SIH inhibited induction of *HAMP* by either BMP6 or IL-6 (Fig 2B), even though DFO did not significantly antagonize the effects of IL-6 (Fig 1A). SIH acts fast and is a cell permeable iron chelator [35]; by contrast, DFO is a slow-acting chelator that enters cells via fluid-phase endocytosis [36].

## Iron inhibits basal and BMP6-mediated SMAD signaling to hepcidin in Huh7 cells in a dose-dependent manner

The SMAD pathway is a central node for signaling to hepcidin in response to iron, inflammatory stimuli and other cues. Having established that iron overload suppresses BMP6-mediated hepcidin induction in Huh7 cells, we performed dose-curve experiments to determine the minimal inhibitory iron concentrations. The data in Fig 3A show that FAC inhibits BMP6-dependent *HAMP* mRNA expression even at low (2–5 μM) doses. Similar results were obtained with *ID1* mRNA (Fig 3B), while inhibition of BMP6-dependent SMAD5 phosphorylation was evident at 50 μM FAC (Fig 3C).

Interestingly, FAC also inhibited basal *HAMP* (Fig 3D) and *ID1* (Fig 3E) mRNAs in a dose-dependent manner, which is indicative of inhibition in basal SMAD signaling. The data in Fig 3F show that *GCLC* mRNA was very sensitive to FAC treatment and was maximally induced

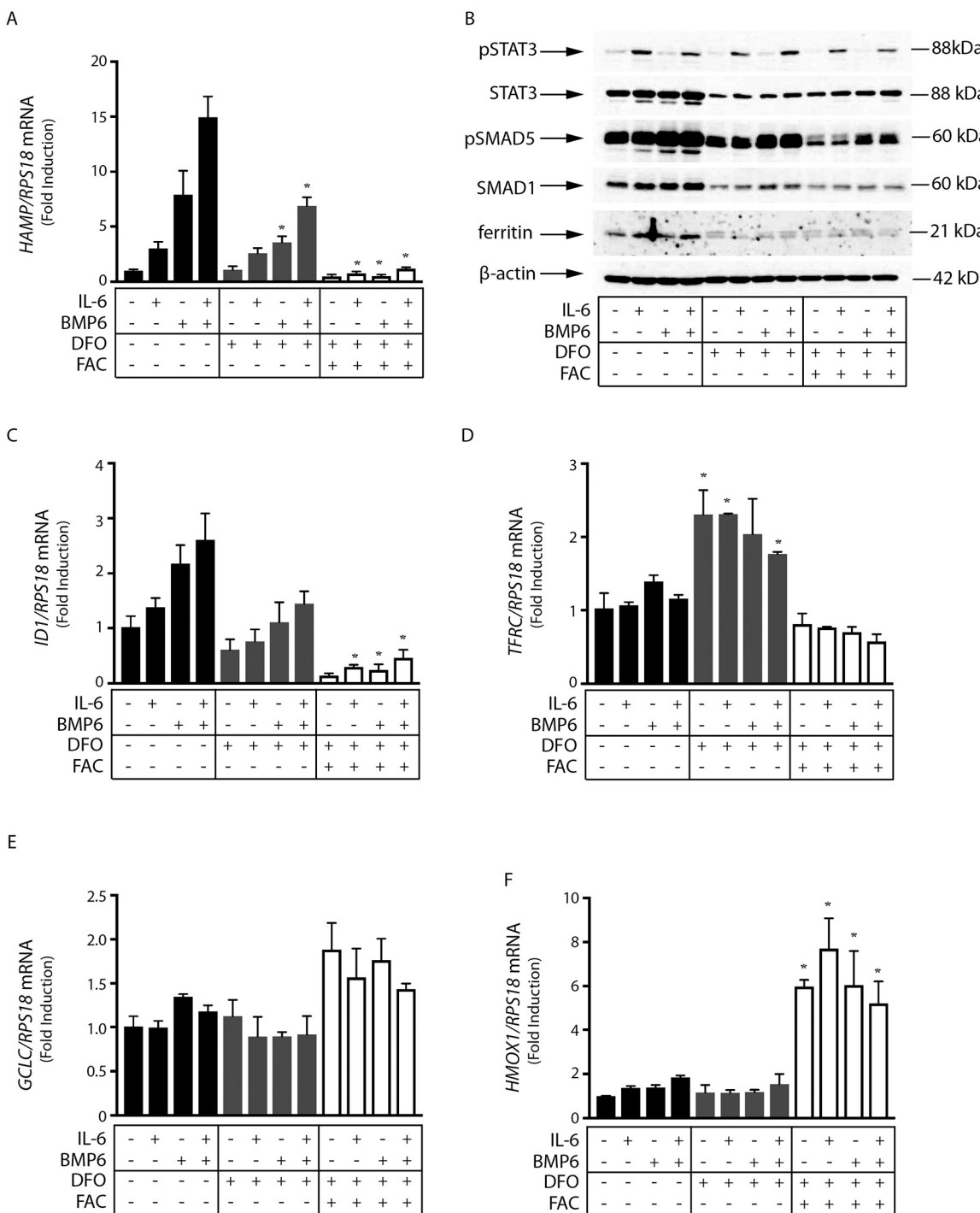

**Fig 1. Iron overload inhibits SMAD and STAT signaling downstream of BMP6 and IL-6 in Huh7 hepatoma cells.** Huh 7 cells were treated with 100 μM DFO for 18 hours before washing and supplementation with 50 μM FAC. After 2 hours, cells were treated with 20 ng/ml human IL-6, 5 ng/ml BMP6, or both over 4 hours. (A) qPCR analysis of *HAMP* mRNA. (B) Western blot analysis of pSTAT3, STAT3, pSMAD5, SMAD5, ferritin and β-actin. (C-F) qPCR analysis of *ID1*, *TFRC*, *GCLC* and *HMOX1* mRNAs. All data in graphs are presented as the mean ± SEM from three independent experiments for (A) and two independent experiments for (B-F). Statistical analysis was performed by one-way ANOVA. Statistically significant differences ($p < 0.05$) across iron treatments (compared to respective values from iron-unperturbed cells shown in black bars) are indicated by *.

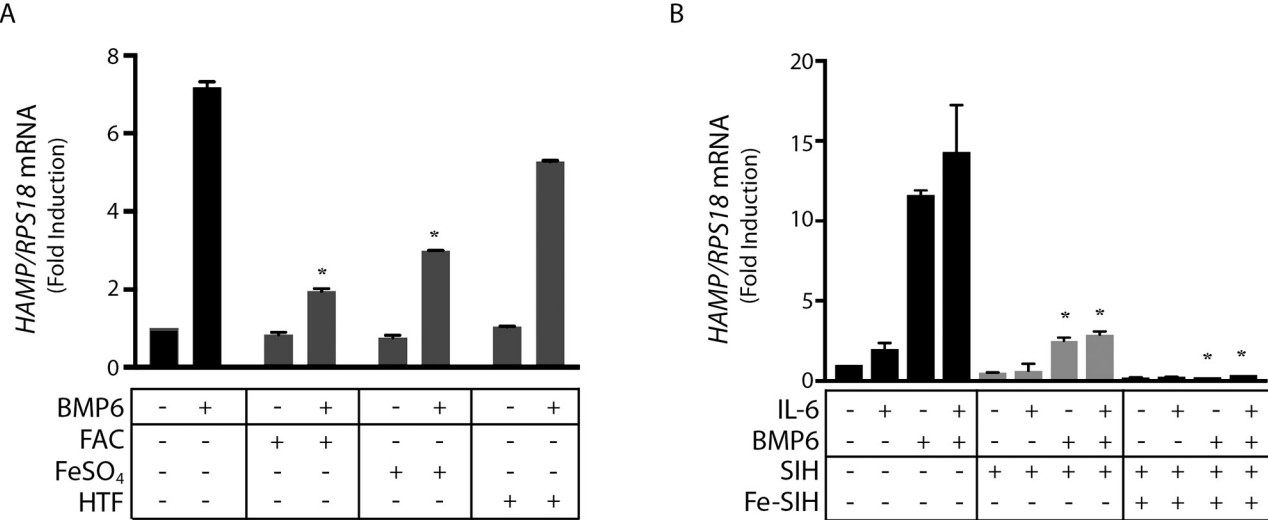

**Fig 2. Iron salts but not transferrin-bound iron inhibit hepcidin induction by BMP6 and/or IL-6 in Huh7 hepatoma cells.** (A) Huh 7 cells were pretreated with either 50 μM FAC, 50 μM FeSO$_4$, or 30 μM holo-transferrin (HTF) for 2 hours and then treated with BMP6 for 4 hours; *HAMP* mRNA was measured by qPCR. (B) Huh 7 cells were treated with 100 μM SIH for 18 hours before washing and supplementation with 50 Fe-SIH. After 2 hours, cells were treated with 20 ng/ml IL-6, 5ng/ml BMP6, or both over 4 hours. *HAMP* mRNA was measured by qPCR. All data in graphs are presented as the mean ± SEM from three independent experiments. Statistical analysis was performed by one-way ANOVA. Statistically significant differences (p<0.05) across iron treatments (compared to respective values from iron-unperturbed cells shown in black bars) are indicated by *.

under these experimental conditions independently of the iron dose. This further supports the notion that exposure of Huh7 cells to FAC triggers oxidative stress.

## Iron-dependent inhibition of BMP/SMAD and IL-6/STAT signaling to hepcidin in primary murine hepatocytes

We used primary murine hepatocytes to address whether the sensitivity of hepcidin signaling pathways to iron perturbations is preserved in another cell model from a different species. Iron chelation by DFO did not significantly affect basal or stimulated expression of hepcidin or other markers of Bmp/Smad or IL-6/Stat signaling in this setting (Fig 4). The data in Fig 4A recapitulate the potent inhibitory effects of FAC on *Hamp* mRNA induction via BMP6 and/or IL-6. Moreover, FAC inhibited basal, as well as BMP6- and IL-6-dependent phosphorylation of Smad5 and Stat3, respectively (Fig 4B). In line with these findings, FAC drastically suppressed BMP6-mediated induction of *Id1* (Fig 4C) and *Smad7* (Fig 4D) mRNAs, both targets of Bmp/Smad signaling [24]. Additionally, FAC antagonized IL-6-mediated induction of *Socs3* mRNA (Fig 4E), a target of IL-6/Stat signaling [37].

We noted that the induction of *Hamp* mRNA in primary hepatocytes by BMP6 (Fig 4A) was more potent compared to the respective *HAMP* mRNA induction in Huh 7 cells (Fig 1A); this may be related to the different doses used, species-specific variations, and/or differences between primary hepatocytes and hepatoma cell lines. We did not examine the effects of iron perturbations on signaling to *Hamp2*, a second hepcidin-encoding gene that is expressed in rodents and fish, because *Hamp2* does not appear to significantly affect systemic iron homeostasis [38, 39]. Furthermore, *Hamp2* expression does not respond to inflammatory stimuli [40].

As expected, exposure of primary murine hepatocytes to the iron chelator DFO elicited strong induction of *Tfrc* mRNA, while subsequent administration of FAC neutralized this response (Fig 4F). By contrast, the FAC treatment failed to induce ferritin, or even neutralize

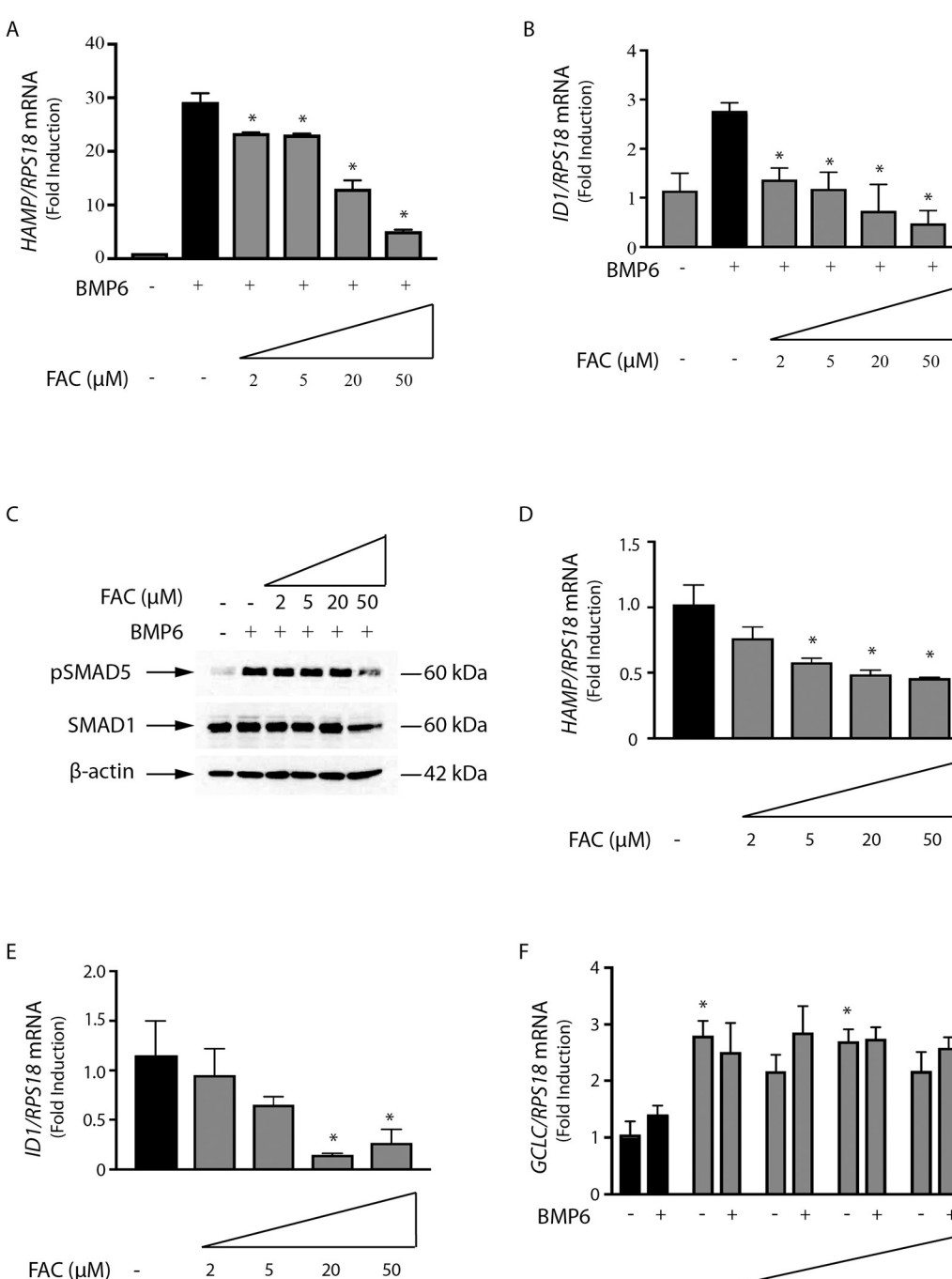

**Fig 3. Dose-dependent inhibitory effects of iron on basal and BMP6-mediated SMAD signaling and hepcidin expression in Huh7 hepatoma cells.** Huh7 cells were treated with increasing doses of FAC over 2 hours. When indicated, the experiment was either terminated or the cells were washed and further incubated with 25 ng/ml BMP6 (A and C) or 5 ng/ml BMP6 (B, D-F) for 4 hours. (A and B) qPCR analysis of *HAMP* and *ID1* mRNAs. (C) Western blot analysis of pSMAD5, SMAD5 and β-actin. (D-F) qPCR analysis of *HAMP*, *ID1* and *GCLC* mRNAs. All data in graphs are presented as the mean ± SEM from three independent experiments. Statistical analysis was performed by one-way ANOVA (A, B, D and E) two-way ANOVA (F). Statistically significant differences (p<0.05) across iron treatments (compared to respective values from iron-unperturbed cells shown in black bars) are directly shown or indicated by *.

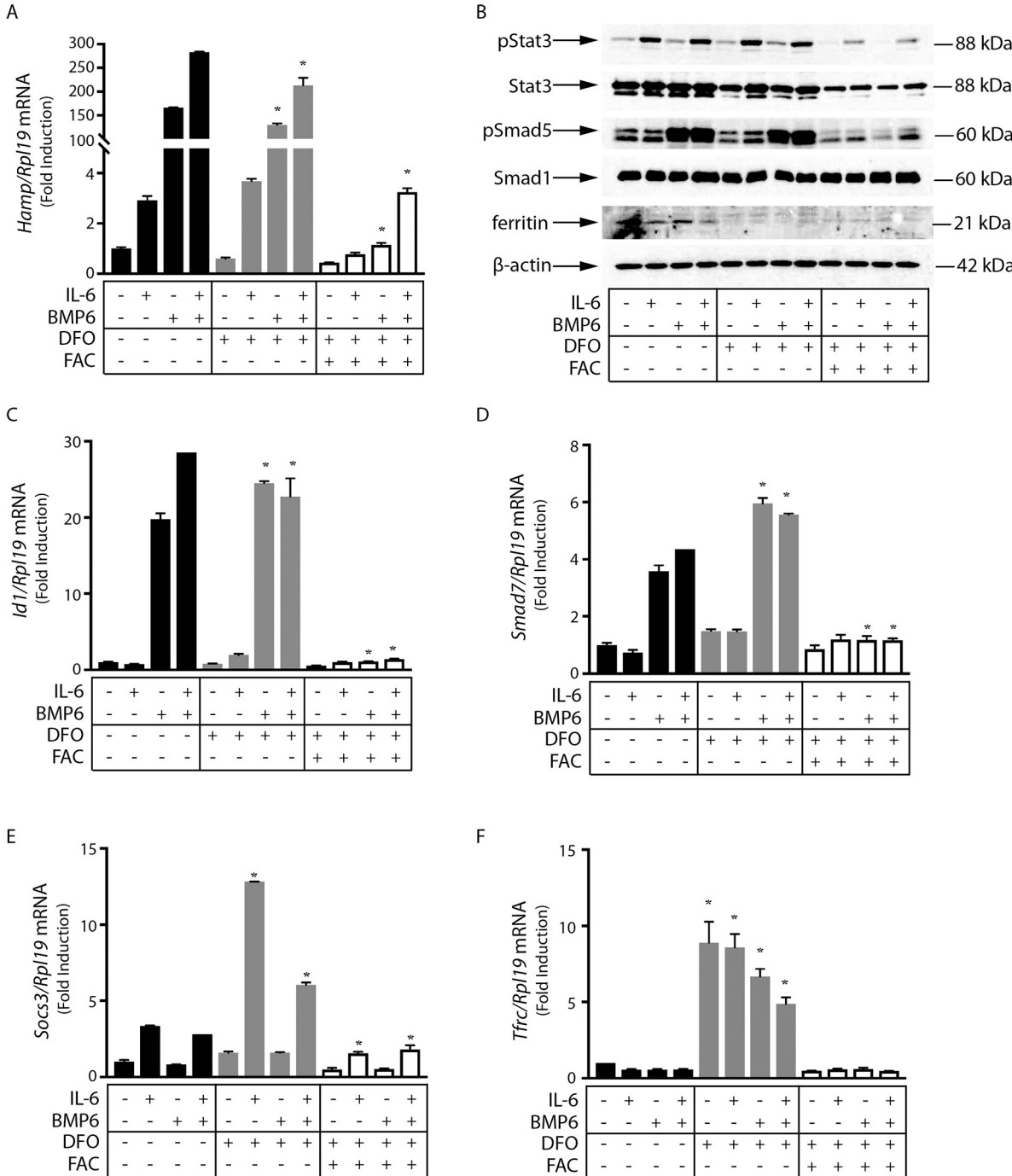

**Fig 4. Iron overload inhibits Smad and Stat signaling downstream of BMP6 and IL-6 in primary murine hepatocytes.** Primary murine hepatocytes were maintained in serum-free William's Medium E over the course of experiments. The cells were pretreated with 100 μM DFO for 18 hours and then washed and supplemented with 50 μM FAC over 2 hours. The cells were then treated with 20 ng/ml murine IL-6, 25 ng/ml BMP6, or both over 4 hours. (A) qPCR analysis of *Hamp* mRNA. (B) Western blotting of pSTAT3, STAT3, pSMAD5, SMAD1, ferritin, and β-actin. (C-F) qPCR analysis of *Id1*, *Smad7*, *Socs3* and *Tfrc* mRNAs. All data in graphs are presented as the mean ± SEM from two independent experiments. Statistical analysis was performed by one-way ANOVA. Statistically significant differences (p<0.05) across iron treatments (compared to respective values from iron-unperturbed cells shown in black bars) are indicated by *.

its suppression by DFO (Fig 2B). Thus, under these experimental conditions, excess iron was apparently not scavenged and remained redox-active.

## Conclusions

We show herein that pharmacological iron chelation slightly impaired hepcidin induction by BMP6 and/or IL-6 primarily in human Huh7 hepatoma cells. Nevertheless, iron supplementation completely abolished BMP6- and/or IL-6-mediated induction of hepcidin in Huh7 cells and primary murine hepatocytes. None of these treatments affected cell viability. The data in Figs 1 and 4 reveal that cellular iron inhibits both BMP/SMAD and JAK/STAT signaling pathways. Considering the relatively short time frame of the experiments that did not allow accumulation of the iron storage protein ferritin, and consequently iron detoxification, we speculate that the inhibitory effects are caused by iron-induced oxidative stress. Experimental support is provided by the induction of *GCLC* and *HMOX1* mRNAs, markers of oxidative stress [32–34], in cells exposed to FAC (Figs 1 and 2). Our findings are consistent with the known sensitivity of both BMP/SMAD and JAK/STAT signaling pathways to oxidants [41, 42], and the previously described suppression of hepcidin in iron-loaded [14, 18, 19] and iron-loaded/IL-6-treated [43] cultured hepatic cells. Moreover, they provide evidence that the iron-mediated suppression of hepcidin in cultured cells is caused by inhibition of basal SMAD signaling. The in vivo relevance of these findings remains to be explored.

## Supporting information

**S1 Fig. Prolonged treatment with FAC increases ferritin steady-state levels.** Huh7 cells were treated with either 100 μM DFO for 18 hours or 50 μM FAC for the indicated time intervals. Ferritin and β-actin expression in cell lysates were assessed by Western blotting. Western data are representative of two independent experiments.
(TIF)

**S1 Table. List of primers used for qPCR.**
(DOCX)

**S1 Dataset.**
(XLSX)

**S1 Raw images.**
(PDF)

## Author Contributions

**Conceptualization:** Edouard Charlebois, Kostas Pantopoulos.

**Formal analysis:** Edouard Charlebois.

**Investigation:** Edouard Charlebois.

**Supervision:** Kostas Pantopoulos.

**Writing – review & editing:** Kostas Pantopoulos.

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
