## [Decision Letter · Decision Letter 0]

12 Jan 2021

PONE-D-20-37619

Iron overload inhibits BMP/SMAD and IL-6/STAT3 signaling to hepcidin in cultured hepatocytes

PLOS ONE

Dear Dr. Pantopoulos,

Thank you for submitting your manuscript to PLOS ONE. After careful consideration, we feel that it has merit but does not fully meet PLOS ONE’s publication criteria as it currently stands. Therefore, we invite you to submit a revised version of the manuscript that addresses the points raised during the review process.

As you can see, both reviewers evaluated your submission as being potentially interesting but also requested additional information that should further strengthen your findings.

We look forward to receiving your revised manuscript.

Kind regards,

Pavel Strnad

Academic Editor

PLOS ONE

Journal Requirements:

Reviewers' comments:

Reviewer's Responses to Questions

**Comments to the Author**

1. Is the manuscript technically sound, and do the data support the conclusions?

Reviewer #1: Partly

Reviewer #2: Partly

2. Has the statistical analysis been performed appropriately and rigorously? 

Reviewer #1: Yes

Reviewer #2: No

3. Have the authors made all data underlying the findings in their manuscript fully available?

Reviewer #1: No

Reviewer #2: No

4. Is the manuscript presented in an intelligible fashion and written in standard English?

Reviewer #1: Yes

Reviewer #2: Yes

5. Review Comments to the Author

Reviewer #1: The manuscript describes an interesting topics; however, substantial additional information needs to be added.

Most importantly, the PLoS Data Policy requires that all data from the experiments should be available. This is definitely not the case. The main information from the manuscript, i.e. the expression of hepcidin, is given only as fold induction. Please include, as Supplementary tables, the actual CT values for the reference gene and hepcidin for the figures.

Since the actal CTs are not mentioned in the manuscript, it is difficult to judge whether the results could be influenced by genomic DNA contamination. The methods do not describe a DNAse treatment step. Did the authors perhaps use the RNeasy PLUS kits, which include a gDNA removal step? Or were the CTs sufficiently high to exclude the effect of gDNA contamination?

The immunoblots do not contain the kilodalton markers (!). Also, please include, as supplementary information, the whole western blot membranes. This will allow the reader to judge the specifity of the antibodies used.

Some of the findings reported in the study are surprising, and should probably be explained. First of all, the authors use ferric amonium citrate for the experiments. The orthodox view is that only ferrous iron is able to pass the cell membrane, with the help of DMT1, ZIP14 etc. Do the authors have a suggestion how ferric iron from FAC enters the cells?

The manuscript does not report the effect of the various treatments on cell viability. Were these in all cases insignificant?

DFO is clearly able to enter the cells and influence TFRC expression; however, on line six of page 7 the authors state that DFO is cell-impermeable. Please explain.

FAC did not increase ferritin synthesis - the authors explanation for this fact is a short incubation period. Did the authors check whether longer incubations induced ferritin protein? This would confirm that iron from FAC actually enters the cells.

Reviewer #2: The manuscript by Edouard Charlebois and Kostas Pantopoulos describing the role of BMP6 and IL6 signaling that activate Hepcidin in cell culture represent the narrow but important field of research. The study is mainly a repetition of already published data with minor new extension.

Major points:

1) The study is based mainly on RT-PCR technic with use of nonvalidated self designed primers which may but doesn't have to recognize specific genes of interest with sensitive concentration dependency.

2) The first half of the paper represents the data from HuH7 cells which is well known human derived hepatic carcinoma cell line. As a comparison to that data authors isolated primary hepatocytes from wild type C57BL/6 mice and repeated the key experiments. Here one have to keep in mind that mice have 2 genes of hepcidin (Hamp1 and Hamp2) that regulate iron metabolism in mice.

3) Authors do not show the statistical details of data evaluation. Please provide the details of experiments such as biological replicates and statistics used for each of the comparison. Additionally from the figures its not clear what authors consider significant, they put the "*" but do not discuss in the legend or elsewhere to what group they compared.

4) The authors should show experimental evidence why they chose that specific time course for each of the treatment.

5) From the results it seems that HuH7 cells and mouse primary hepatocytes display different mechanism of Hepcidin activation. The authors should discuss it.

6) Through the figures some controls are missing for instance on Fig 1C no control for DFO w/o other treatments. Please make sure that all controls are presented.

7) It is not clear why authors used RT-PCR method to detect oxidative stress caused by iron. Also HMOX1 is not a vary good marker for oxidative stress.... Please provide robust prove of oxidative stress such as 4HNE or other similar. In case of positive result in cells exposed to FAC, provide a clear discussion on oxidative stress.

6. PLOS authors have the option to publish the peer review history of their article (what does this mean?). If published, this will include your full peer review and any attached files.

Reviewer #1: **Yes: **Jan Krijt

Reviewer #2: No

---

## [Author Response · Author response to Decision Letter 0]

25 May 2021

We thank both reviewers for their constructive comments. A point-by-point response to their comments is provided below:

Reviewer 1

1) The manuscript describes an interesting topics; however, substantial additional information needs to be added.

Most importantly, the PLoS Data Policy requires that all data from the experiments should be available. This is definitely not the case. The main information from the manuscript, i.e. the expression of hepcidin, is given only as fold induction. Please include, as Supplementary tables, the actual CT values for the reference gene and hepcidin for the figures.

Since the actual CTs are not mentioned in the manuscript, it is difficult to judge whether the results could be influenced by genomic DNA contamination. The methods do not describe a DNAse treatment step. Did the authors perhaps use the RNeasy PLUS kits, which include a gDNA removal step? Or were the CTs sufficiently high to exclude the effect of gDNA contamination? 

Excel files containing all the raw data for qPCRs have been included with this revision. In general, CT values were much higher than their non-template controls, and melt curves demonstrated clear single peaks. Primers are tested on samples to generate efficiency curves which all yield 90-100% efficiency. Melt curves and primer efficiencies are available upon request. We used RNeasy mini kits and the Onescript plus reverse transcriptase kit which does not remove gDNA. However, qPCR results show no indication of gDNA contamination due to their reproducibility. 

2) The immunoblots do not contain the kilodalton markers (!). Also, please include, as supplementary information, the whole western blot membranes. This will allow the reader to judge the specifity of the antibodies used.

Whole western blot membranes are now included as supplemental information. Kilodalton markers have been added to the Figures. 

3) Some of the findings reported in the study are surprising, and should probably be explained. First of all, the authors use ferric amonium citrate for the experiments. The orthodox view is that only ferrous iron is able to pass the cell membrane, with the help of DMT1, ZIP14 etc. Do the authors have a suggestion how ferric iron from FAC enters the cells?

This issue is discussed in lines 129-131 of the revised manuscript.

4) The manuscript does not report the effect of the various treatments on cell viability. Were these in all cases insignificant?

We report in the revised manuscript that none of the treatments affected cell viability (line 215).

5) DFO is clearly able to enter the cells and influence TFRC expression; however, on line six of page 7 the authors state that DFO is cell-impermeable. Please explain.

This point is addressed in lines 158-160 of the revised manuscript.

6) FAC did not increase ferritin synthesis - the authors explanation for this fact is a short incubation period. Did the authors check whether longer incubations induced ferritin protein? This would confirm that iron from FAC actually enters the cells.

This important issue is addressed by a new experiment that is shown in Fig. S1. The data indeed confirm that FAC enters the cells and together with previous kinetic data on FAC uptake by hepatocytes (discussed in lines 145-146), they support our interpretation for early accumulation of unshielded iron within the cells.

Reviewer 2

1) The study is based mainly on RT-PCR technic with use of non-validated self-designed primers which may but doesn't have to recognize specific genes of interest with sensitive concentration dependency.

Although the primers are mostly self-designed, they have indeed been validated through the NIH’s primer-BLAST tool. We have performed extensive analysis of melt curves as well as performed experiments to determine primer efficiency. Only primers with efficiency between 90% and 100% are used and primers with single melt curve peaks. Data are available upon request. Furthermore, primers which have not been previously published (mainly human SOCS3, GCLC, and ID1) were used in experiments with multiple replicates and produced consistent and reproducible results.

2) The first half of the paper represents the data from HuH7 cells which is well known human derived hepatic carcinoma cell line. As a comparison to that data authors isolated primary hepatocytes from wild type C57BL/6 mice and repeated the key experiments. Here one have to keep in mind that mice have 2 genes of hepcidin (Hamp1 and Hamp2) that regulate iron metabolism in mice.

This issue has been addressed in lines 195-204 of the revised manuscript.

3) Authors do not show the statistical details of data evaluation. Please provide the details of experiments such as biological replicates and statistics used for each of the comparison. Additionally from the figures its not clear what authors consider significant, they put the "*" but do not discuss in the legend or elsewhere to what group they compared.

Biological replicates are indicated in the figure legends. We have updated the Statistical Analysis in the Materials & Methods section. We have modified Figures 2-4 and indicate the reference point for statistical significance with black bars. The Figure legends have also been updated for clarity.

4) The authors should show experimental evidence why they chose that specific time course for each of the treatment.

The choice of doses and time points in the various treatments was based on previous studies (lines 121-122).

5) From the results it seems that HuH7 cells and mouse primary hepatocytes display different mechanism of Hepcidin activation. The authors should discuss it.

This is discussed in lines 193-195.

6) Through the figures some controls are missing for instance on Fig 1C no control for DFO w/o other treatments. Please make sure that all controls are presented.

The DFO alone control is included in all Figures (see fifth lanes). If the reviewer meant FAC with no other treatments, these effects are observed in experiments with FAC dose curves (Fig. 3) where the inhibition of hepcidin is also observed even in the absence of DFO pre-treatment.

7) It is not clear why authors used RT-PCR method to detect oxidative stress caused by iron. Also HMOX1 is not a very good marker for oxidative stress.... Please provide robust prove of oxidative stress such as 4HNE or other similar. In case of positive result in cells exposed to FAC, provide a clear discussion on oxidative stress.

We attempted to detect oxidative stress by using the TBARS assay but could not find significant differences (data not shown). We respectfully disagree that HMOX1 is not a very good marker. Experimental data suggest that HMOX1 and GCLC are sensitive oxidative stress markers as both genes are transcriptionally regulated by Nrf2, a major redox sensor. We added another reference (#33) supporting this statement. We argue that the induction of HMOX1 and GCLC observed in our experiments are early signs of oxidative stress due to accumulation of unshielded iron, which has not yet manifested to lipid peroxidation and membrane damage.

---

## [Decision Letter · Decision Letter 1]

7 Jun 2021

Iron overload inhibits BMP/SMAD and IL-6/STAT3 signaling to hepcidin in cultured hepatocytes

PONE-D-20-37619R1

Dear Dr. Pantopoulos,

We’re pleased to inform you that your manuscript has been judged scientifically suitable for publication and will be formally accepted for publication once it meets all outstanding technical requirements.

Kind regards,

Pavel Strnad

Academic Editor

PLOS ONE

Additional Editor Comments (optional):

Reviewers' comments:

Reviewer's Responses to Questions

**Comments to the Author**

1. If the authors have adequately addressed your comments raised in a previous round of review and you feel that this manuscript is now acceptable for publication, you may indicate that here to bypass the “Comments to the Author” section, enter your conflict of interest statement in the “Confidential to Editor” section, and submit your "Accept" recommendation.

Reviewer #1: All comments have been addressed

2. Is the manuscript technically sound, and do the data support the conclusions?

Reviewer #1: (No Response)

3. Has the statistical analysis been performed appropriately and rigorously? 

Reviewer #1: (No Response)

4. Have the authors made all data underlying the findings in their manuscript fully available?

Reviewer #1: (No Response)

5. Is the manuscript presented in an intelligible fashion and written in standard English?

Reviewer #1: (No Response)

6. Review Comments to the Author

Reviewer #1: (No Response)

7. PLOS authors have the option to publish the peer review history of their article (what does this mean?). If published, this will include your full peer review and any attached files.

Reviewer #1: **Yes: **Jan Krijt

---

## [Editor Report · Acceptance letter]

14 Jun 2021

PONE-D-20-37619R1 

Iron overload inhibits BMP/SMAD and IL-6/STAT3 signaling to hepcidin in cultured hepatocytes 

Dear Dr. Pantopoulos:

I'm pleased to inform you that your manuscript has been deemed suitable for publication in PLOS ONE. Congratulations! Your manuscript is now with our production department. 

Kind regards, 

on behalf of

Dr. Pavel Strnad 

Academic Editor

PLOS ONE